# Long-Lasting Insecticide-Treated Nets Combined or Not with Indoor Residual Spraying May Not Be Sufficient to Eliminate Malaria: A Case-Control Study, Benin, West Africa

**DOI:** 10.3390/tropicalmed8100475

**Published:** 2023-10-18

**Authors:** Barikissou G. Damien, Thomas Kesteman, Gatien A. Dossou-Yovo, Amal Dahounto, Marie-Claire Henry, Christophe Rogier, Franck Remoué

**Affiliations:** 1MIVEGEC (Maladies Infectieuses et Vecteurs: Ecologie, Génétique, Evolution et Contrôle), Université de Montpellier, CNRS, IRD, 911 Avenue Agropolis BP 64501, 34394 Montpellier, France; s.dossou-yovo@ch-ouestguyane.fr (G.A.D.-Y.); amal.dahounto@gmail.com (A.D.); marieclairechantal.henry@gmail.com (M.-C.H.); franck.remoue@ird.fr (F.R.); 2Centre de Recherche Entomologique de Cotonou (CREC), Cotonou 06 BP 2604, Benin; 3Malaria Research Unit, Institute Pasteur de Madagascar, BP 1274 Avaradoha, Antananarivo 101, Madagascar; tkesteman@oucru.org (T.K.); christophe.rogier@primumvitare.com (C.R.); 4Primum Vitare, 118 Avenue Félix Faure, 75015 Paris, France

**Keywords:** malaria, infection, uncomplicated clinical cases, LLINs, IRS, effectiveness, Benin

## Abstract

In sub-Saharan Africa, despite the implementation of multiple control interventions, the prevalence of malaria infection and clinical cases remains high. The primary tool for vector control against malaria in this region is the use of long-lasting insecticide-treated nets (LLINs) combined or not with indoor residual spraying (IRS) to achieve a synergistic effect in protection. The objective of this study was to assess the effectiveness of LLINs, with or without IRS, protected against *Plasmodium falciparum* infection and uncomplicated clinical cases (UCC) of malaria in Benin. A case-control study was conducted, encompassing all age groups, in the urban area of Djougou and the rural area of Cobly. A cross-sectional survey was conducted that included 2080 individuals in the urban area and 2770 individuals in the rural area. In the urban area, sleeping under LLINs did not confer significant protection against malaria infection and UCC when compared to no intervention. However, certain neighbourhoods benefited from a notable reduction in infection rates ranging from 65% to 85%. In the rural area, the use of LLINs alone, IRS alone, or their combination did not provide additional protection compared to no intervention. IRS alone and LLINs combined with IRS provided 61% and 65% protection against malaria infection, respectively, compared to LLINs alone. The effectiveness of IRS alone and LLINs combined with IRS against UCC was 52% and 54%, respectively, when compared to LLINs alone. In both urban and rural areas, the use of LLINs alone, IRS alone, and their combination did not demonstrate significant individual protection against malaria infection and clinical cases when compared to no intervention. In the conditions of this study, LLINs combined or not with IRS are not effective enough to eliminate malaria. In addition to the interventions, this study identified factors associated with malaria in Benin as housing design, neglected social groups like gender-marginalised individuals and adolescents, and socio-economic conditions acting as barriers to effective malaria prevention. Addressing these factors is crucial in order to facilitate malaria elimination efforts in sub-Saharan Africa.

## 1. Introduction

Despite a significant decline in infections and clinical cases observed between 2000 and 2015, which has been attributed to long-lasting insecticide-treated nets (LLINs) and indoor residual spraying (IRS), especially in settings with moderate to high transmission, the malaria burden is increasing again in some sub-Saharan African countries [1,2].

LLINs are bed nets treated with insecticides, mainly pyrethroids, designed to prevent mosquitoes from biting people indoors and reduce the mosquito population. The insecticides also repel mosquitoes, thereby limiting their entry into households. The World Health Organization (WHO) encourages the continued use of LLINs for malaria prevention. It recommends that all sleeping units should be covered by LLINs, and the entire population living in endemic areas should sleep under LLINs every night [3,4].

IRS is implemented as a complementary strategy if needed and when the necessary material and financial conditions are met [1,2]. It involves the periodic application of chemical insecticides to the interior walls of dwellings, thereby reducing the number of resting sites for mosquitoes. The rationale behind combining these two interventions relies on optimising their effectiveness through synergistic actions. In recent years, the high level of resistance of mosquito vectors to pyrethroids used in impregnated nets has strongly encouraged the concept of combining both interventions in national malaria control programmes (NMCP) [5].

Typically, non-pyrethroid insecticides are used for IRS. The underlying assumption is that non-pyrethroid IRS can induce an additional impact in cases where mosquitoes are resistant to pyrethroids on LLINs but remain susceptible to non-pyrethroids delivered through IRS. This approach is also regarded as a resistance management strategy, even though pyrethroid resistance levels have already reached their maximum thresholds. However, all these assumptions need periodic evaluation in real-world conditions [6].

In the realm of health interventions, theoretical efficacy often differs from the observed real-world effectiveness. The level of protection conferred by vector control interventions is contingent on the specific context and various local factors such as vector and human behaviour, the effectiveness of health systems, and the level of coverage achieved by the intervention [7,8,9]. A longitudinal study conducted in the peri-urban area of Cotonou in southern Benin has revealed that the use of bed nets is associated with only a 20% reduction in malaria infection [10]. A phenomenon known as exophily and exophagy was also observed, a consequence of increasing prevalence of mosquitoes resting and biting outside the home. This behavioural shift poses a challenge to implementing insecticide-treated bed nets and IRS that primarily target indoor mosquito populations [11]. The trend of mosquitoes biting earlier in the day has also been documented in *Anopheles* spp. mosquitoes in West Africa. This change in biting behaviour could have implications for malaria prevention and control strategies, potentially impacting the effectiveness of interventions that are designed to target mosquitoes during specific times of the day or night [11]. Extending outdoor staying or sleeping without bed nets due to heat can increase the probability of human–vector contact and malaria infection. The environmental context, whether urban or rural, is often very different. It is therefore essential to analyse the data, taking account of spatial micro-stratification. While it is well-recognised that the malaria burden can vary across different age groups, it seems that less attention is given to the disease among older children and adolescents (6–15 years) compared to children under five years old. Recognising and addressing the specific challenges and vulnerabilities faced by different age groups is crucial for effective malaria control and prevention strategies [12,13]. 

In order to address these challenges, this study was conducted to evaluate the effectiveness of major vector control interventions, such as LLINs used alone or in combination with IRS, in reducing *Plasmodium falciparum* infections and uncomplicated clinical cases (UCC) in the overall population, including both urban and rural areas, compared to no intervention. The findings of this study can provide valuable insights for NMCPs in updating their strategies and allocating resources at the local level. Through a comprehensive understanding of the contributions of different interventions, governments via NMCPs can make informed decisions to enhance malaria control efforts.

## 2. Materials and Methods

### 2.1. Study Area

This study was conducted in 2014 in two communities located in the north-west region of Benin: Djougou, an urban area, and Cobly, a rural area in Benin. Benin is a lower-middle-income country located in West Africa.

In 2021, approximately 49.0% of the country’s population resided in urban areas [14]. Children aged under 14 years old accounted for 42.6% of the total population during the period of 2017 to 2018 [15]. The informal economy was estimated to represent 55.6% of economic activities in Benin. This indicates a substantial portion of economic activities occurring outside the formal sector, often characterised by informal employment, unregistered businesses, and a lack of social protection mechanisms [16]. The 2017–2018 Demographic Health Survey revealed a relatively low level of education among adults in Benin. The survey indicated that approximately 32% of men and 55% of women had never attended school, highlighting the educational challenges faced by a significant portion of the adult population, particularly among women, with respect to accessing formal education [15].

In both Djougou and Cobly, the dry season typically spans from November to April, followed by a rainy season from May to October. The average temperature in the area hovers around 27 °C, with fluctuations ranging between 25 °C and 35 °C. Djougou, located approximately 450 km away from Cotonou, the economic capital of Benin, covers an area of 3926 square kilometres [17]. The size of the population was 294,056 inhabitants in 2023 [18]. The rural district of Cobly covers an area of 825 square kilometres [19]. The size of the population was estimated at 75,558 inhabitants in 2023 [18].

Malaria remains “hyperendemic” in the north-west departments of Benin, specifically in Donga and Atacora. Hyperendemic refers to a high and persistent level of malaria transmission in an area, with a prevalence of *P. falciparum* infection ranging between 50% and 75% [20]. During the study period, entomological data were also collected revealing an entomological inoculation rate of five infective bites/person/month on average in Djougou and two infective bites/person/month in Cobly. The allelic frequency of the kdr mutation was very high in both areas: 95% in Djougou and 100% in Cobly [21].

In Benin, several strategies related to the intensification of malaria control were defined by the NMCP during different periods, namely, from 2006 to 2010, 2011 to 2018, and 2017 to 2021. These strategies were primarily based on the widespread deployment of LLINs, IRS implementation in selected health districts, intermittent preventive treatment in pregnant women with sulfadoxine-pyrimethamine (IPTp-SP), and treatment with artemisinin-based combination therapy (ACT). Recently, seasonal malaria chemoprevention has been implemented in the northern regions of Benin, where malaria transmission is highly seasonal. The introduction of a malaria vaccine (RTS, S) for children under 5 years old is currently being considered. The initial nation-wide campaign for LLIN distribution took place in 2007, targeting children under 5 years old and pregnant women. Subsequently, universal distributions of LLINs were conducted nationwide in 2011, and repeated in 2014, 2017, and 2020.

A notable strength of IRS implementation in Benin was the country’s ability to carry out large-scale operations, resulting in the treatment of over 80% of houses in the targeted regions. In Benin, IRS campaigns were conducted once a year, typically in May before the start of the rainy season. As per the government criteria based on epidemiological indicators, while the Cobly area benefited from IRS, the Djougou area was not eligible for IRS interventions during that time. In Cobly, the first edition of IRS was implemented in 2011 using the insecticide bendiocarb [22]. In 2014, pirimiphos-methyl 300 CS (PM 300 CS) insecticide was used for IRS exhibiting and its residual activity lasted 4–5 months on both mud and cement walls [22,23,24]. PM 300 CS induced more than 80% mortality in susceptible “Kisumu” *Anopheles gambiae* after 24 h, 48 h, and 72 h of exposure [24]. Other aspects of the methodology used for implementing IRS in Benin is described elsewhere [22,23,24].

### 2.2. Material and Methods

#### 2.2.1. Study Design and Sampling

A case-control study was conducted to evaluate the effectiveness of LLINs and IRS against *P. falciparum* infections and UCC. Cases and controls were selected from a cross-sectional survey undertaken from July to August in Djougou and in September in Cobly in 2014, i.e., during the rainy season.

Each village/neighbourhood is composed of one or more hamlets. Households were selected using a random approach. The sampling procedure was conducted per hamlet. The number of households to be included per hamlet was predetermined. A random point on a map of the hamlet was chosen to indicate the initial household, and surveyors then proceeded to visit the second household on the right when stepping out of the previous household. Only one random point was chosen per hamlet except if an impassable obstacle was found before the end of the street. In this case, a new random point of mapping was chosen. The study aimed to include a minimum of 50 households in each of the eight villages in rural areas and eight neighbourhoods in urban areas, with at least 250 individuals across all age groups. The sample size for the case-control study assumed an odds ratio (OR) of 0.5, a proportion of people exposed to LLINs (the intervention) of 50%, a statistical power of 80%, and a probability of Type I error of 5% [25]. The calculation showed that a minimum of 100 UCCs (1 case/1 control) and 148 infection cases (1 case/3 controls) at least were needed. 

The inclusion criteria of the cross-sectional study were as follows: (i) living in the study area for at least 3 months, (ii) being recruited from the community during the reference period, (iii) informed consent being obtained from the participants, particularly for the collection of blood samples, and (iv) being able to answer questions and actively participate in the study. In the second step, cases of infection, UCCs, and their respective controls were selected following the definitions provided below.

#### 2.2.2. Data Collection and Analysis

The data collection process consisted of two steps:

In the first step, socio-anthropologists conducted household visits and collected individual and household-level data using a standardised questionnaire, including demographic information, socio-economic status, vector control interventions, and medicines used for malaria treatment. Sleeping under LLINs every night was recorded as a binary variable, coded as “Yes” if the individual slept under the LLINs every night during the week preceding the survey and “No” if they slept one to six times or less under LLINs that week. In the Cobly area, where both LLINs and IRS interventions were implemented, the exposition to LLINs alone, IRS alone, and a combination of LLINs and IRS were compared to individuals in the “no intervention” category. Additionally, LLINs use was compared to IRS alone and LLINs combined with IRS.

Additional individual-level data were collected, including the age and gender of each participant, use of any other methods or tools to prevent mosquito bites in addition to LLINs, use of malaria prophylaxis as well as LLINs to prevent malaria, and usual treatment of malaria episodes. Household-level information was also collected, which included the level of education of the head of the household, household size (the number of individuals residing in the household), wealth index (the household’s economic status or wealth), place of residence, respondent’s level of education, closing of openings at night, e.g., doors and windows, and presence of open eaves space between the wall and roof. These variables were collected to capture important factors related to individual behaviours, socio-economic status, and household characteristics that could potentially influence the risk of malaria.

In the second step of the data collection process, parasitological and clinical information was gathered from participants. Trained nurses recorded axillary temperatures and performed diagnostic tests. Fever was defined as a body temperature ≥37.5 °C or a history of fever within 48 h preceding the survey. *P. falciparum* is the most common species of malaria in Benin [26]. Immunochromatographic malaria rapid diagnostic tests (mRDTs) were used to determine the participants’ malaria infection status [25,27]. These mRDTs are designed to detect the presence of *P. falciparum* by identifying the histidine-rich protein 2 antigenemia (HRP2). This information was valuable for assessing the prevalence of malaria infection in the study population [25,28]. However, mRDTs are generally designed to detect recent or acute infections, typically within the past 2 weeks. The CareStart™ mRDT was used.

Malaria UCCs were defined by the presence of fever or a fever history within 48 hours before the survey, coupled with confirmation of a positive thick blood smear specifically indicating the presence of *P. falciparum*. Individuals with fever or a fever history with a negative thick blood smear served as controls. It is important to note that severe clinical cases were excluded from the study as the focus was solely on uncomplicated cases of malaria.

The data analysis employed a logistic regression model to assess the association between interventions and malaria infection or UCC, using SAS version 9.3 (SAS Institute Inc., Cary, NC, USA). The model constantly included several variables, such as age group, gender, and place of residence, to account for their potential influence on the outcomes. Individuals were categorised according to age into five groups: 0–4 years, 5–9 years, 10–14 years, 15–40 years, and 41 years and above. The wealth index was generated using a standardised principal component analysis (PCA) that considered various household characteristics and factors related to wealth status within each of the two sites, including household characteristics such as the household’s goods and possessions (transports, sewing machine, television, radio, health mutual, bank account, the monthly income of the household provider, etc.), the dwelling conditions (walls, rooves, and floors; the presence of a fireplace in the room where the individuals sleep; access to drinking water; the type of fuel; and lighting) [29,30]. The scoring system allowed for the classification of households into five groups representing different levels of wealth status based on ascending order. These categories were labelled as “most poor”, “very poor”, “poor”, “less poor”, and “least poor”.

By incorporating the wealth groups and other covariates into the logistic regression model, the analysis aimed to account for socioeconomic factors possibly confounding the association between the interventions and malaria outcomes. This approach helped to control for possible biases and provided a more accurate assessment of the relationship between LLINs, IRS, and malaria infection/UCC. 

Interactions among variables related to vector control interventions were investigated, including interactions between these variables, interactions with age, and interactions with household characteristics. In addition to age, gender, place of residence, and wealth, variables that significantly modified the odds ratio (OR) were also retained in the final model. Variables without significant effects or that did not contribute to the understanding of the relationship were stepwise eliminated. This process helped to simplify the model and focus on the key variables that had a meaningful impact on the outcomes. ORs and their 95% confidence intervals were estimated to assess the strength and direction of the associations. Protective effectiveness (PEe) was calculated using the formula PEe = 1 − OR to estimate the protection conferred by the malaria vector control intervention in reducing the risk of malaria infection or UCC.

## 3. Results

### 3.1. Study Sites and Population

The study consisted of a total of 18 sites, with 9 sites investigated in each study area. In total, 876 households were included in the study, with 449 households in Djougou and 427 households in Cobly. The prevalence rate of *P. falciparum* infection, as determined by mRDT, was found to be 29% and 53% in Djougou and Cobly, respectively. The prevalence rates were highest among children aged 2 to 9 years old: 36% in Djougou and 70% in Cobly.

#### 3.1.1. Djougou Area 

Of the 2080 participants included in the study, 56% were female and 23% were children under 5 years old. The majority of heads of household (62.1%) were uneducated. Regarding the living conditions, 77.5% of people lived in houses without window screens or nets (Table 1a).

#### 3.1.2. Cobly Area

A total of 2770 individuals participated in the study; among them, 19.5% were aged under five years old and 50.1% were female. The majority (76.0%) of the heads of household were uneducated. Regarding living conditions, almost all (98.8%) of the participants lived in houses without window screens or nets (Table 1b). 

### 3.2. Coverage and Use of Malaria Control Measures

In Djougou, 78% of households had at least one LLIN and 28% of households had one LLIN for two people. In comparison, in Cobly, these indicators were 80% and 35%, respectively. The percentage of individuals who slept under LLINs every night in the week before the survey was 39% in Djougou and 44% in Cobly. However, LLINs use showed a significant decrease with age. In Djougou, the proportions of individuals sleeping under LLINs were 48%, 39%, 27%, and 28% for those aged under 5 years, 5–9 years, 10–14 years, and above 14 years old, respectively. In Cobly, the proportions were 67%, 55%, 48%, and 43% for the corresponding age groups. The use of other tools against mosquito bites, such as mosquito coils, domestic insecticide spray, and smoking with traditional grass, was reported by 86% of individuals in Djougou and 23% in Cobly. In Cobly, 96% of households were covered by IRS. The breakdown of population coverage was as follows: 6% were covered by LLINs only, 14% by IRS only, and 80% by both interventions (LLINs and IRS). 

### 3.3. Protection Conferred by LLINs Combined or Not with IRS

#### 3.3.1. Protection Conferred by the Use of Long-Lasting Insecticide-Treated Nets on Malaria Infection and Uncomplicated Clinical Cases in Djougou, Urban Area

A total of 594 infections and 1495 controls were included. In the multivariate analysis, it was found that the use of LLINs reduced malaria infection in four out of eight villages when considering the interaction with the place of residence and adjusting for age, gender, and wealth index. However, factors such as the education level of the responding individuals, household size, screened windows, open eaves, use of other tools against mosquito bites (such as mosquito coils, aerosol bombs, fumigations with plants and insecticides), and use of anti-malarial drugs were not significantly associated with malaria infection. The risk of malaria infection was higher in the age group of individuals aged 5 to 9 years (OR = 1.70 [1.27–2.28]) and even higher in those aged 10 to 14 years (OR = 2.34 [1.70–3.22]) compared to children aged 0 to 4 years. On the other hand, individuals aged 41 years or older had a lower risk of malaria infection compared to children aged 0 to 4 years (OR = 0.49 [0.31–0.79]). Females were found to be less infected than males (OR = 0.77 [0.63–0.95]). Additionally, the infection rate decreased with an increase in wealth index status (*p* < 0.0001). These results indicate that LLINs use, age, gender, and wealth index were important factors associated with malaria infection.

Similarly, the analysis showed that there was no significant association between UCC of malaria and the use of LLINs (OR = 0.95 [0.67–1.35]) even after adjusting for age, wealth index, open eaves, and place of residence (Table 2b).

#### 3.3.2. Protection Conferred by the Use of Long-Lasting Insecticide-Treated Nets Alone or in Combination with Indoor Residual Spraying on Malaria Infection and Clinical Cases in Cobly, Rural Area

A total of 1140 infected participants and 1133 controls (uninfected) were included in the study. In the multivariate analysis, there was no significant reduction in malaria infection when comparing LLINs alone, IRS alone, and LLINs + IRS to no intervention, after adjusting for factors such as “age”, “gender”, “education level”, and “place of residence”. However, when comparing the different interventions, it was found that the use of IRS alone (OR = 0.39 [0.15–0.99], *p* = 0.0482) or LLINs combined with IRS (OR = 0.35 [0.13–0.89], *p* = 0.0276) resulted in a significant reduction in malaria infection compared to LLINs alone, after adjusting for the aforementioned factors. This indicates that the combination of LLINs and IRS showed a greater protective effect against malaria infection compared to LLINs alone. Interestingly, the analysis also revealed that male individuals had a higher risk of malaria infection compared to females, as indicated by the higher odds ratio (OR) observed in Table 3a.

A total of 267 UCC of malaria and 1906 uninfected controls were included in the study. The multivariate analysis examined the effectiveness of different interventions in reducing UCC, including the use of LLINs alone, IRS alone, and a combination of LLINs and IRS. The results of the analysis showed that there was no significant reduction in UCC when comparing LLINs alone, IRS alone, and LLINs + IRS to no intervention, after adjusting for factors such as “age”, “gender”, “educational level”, “place of residence”, and “open eaves”. This suggests that none of these interventions alone were effective in significantly reducing the risk of UCC in the study population. However, when comparing the different interventions to each other, it was found that a significant reduction in UCC was observed when comparing IRS alone (OR = 0.48 [0.26–0.90], *p* = 0.0010) or LLINs combined with IRS (OR = 0.46 [0.25–0.85], *p* = 0.0142) to LLINs alone, after adjusting for the aforementioned factors. This indicates that IRS alone or a combination of LLINs and IRS has a significant protective effect against UCC compared to LLINs alone. Furthermore, the analysis also revealed that UCC increased with age (*p* < 0.0001) and when eaves were opened (*p* = 0.0017), indicating that these factors were associated with a higher risk of UCC.

## 4. Discussion

The findings emphasise the ongoing challenges in meeting the expectations of populations, decision makers, and granters and final supporters regarding malaria vector control strategies in developing countries. Despite the implementation of interventions such as LLINs and IRS, there is still a need for improved strategies to effectively control malaria. The study conducted in Benin provides valuable insights into the malaria burden, its risk factors, and the benefits and limitations of vector control tools, specifically IRS and LLINs. By examining the data and analysing the association between different variables, the study contributes to a better understanding of the factors influencing malaria transmission and the effectiveness of existing interventions. The findings of this study can help inform decision-making processes and guide the NMCP in addressing persistent and resurging malaria cases in certain localities. The research underscores the importance of ongoing research efforts and the prioritisation of malaria vaccine development as a key strategy recommended by the WHO. 

### 4.1. Long-Lasting Insecticide-Treated Nets and Indoor Residual Spraying Coverage and Use

LLINs and IRS were implemented as primary vector control interventions in northern Benin. The coverage rate of IRS was very high in the region. Additionally, the percentage of households owning at least one LLIN in both rural and urban areas was higher (78–80%) compared to the data recently published by the World Malaria Report (WMR), which reported a global LLINs ownership rate of 65%. However, the percentage of households owning at least one LLIN for every two people is relatively low (28–35%) in both rural and urban areas of Benin. This finding aligns with the global trend where LLINs ownership has increased from 1% in 2000 to 34% in 2020 [2]. The study findings indicate that the use of LLINs in the study area, as well as the global average, is low. The reported data suggest that only 50% of people, including children under 5, actually sleep under a mosquito net in the world. As the study was carried out in the rainy season, it was expected that during periods of increased vector activity and malaria transmission, the use of LLINs and the acceptance of the IRS would be the highest. The highest vector activity during the rainy season should serve as a motivating factor for people to use LLINs and seek protection by sleeping in sprayed huts [31,32,33]. This highlights a low compliance to LLINs even in high transmission periods. These results support the argument that we need to be looking beyond LLINs (at least those treated with pyrethroids) for malaria control.

Contrarily, the high temperatures experienced during the dry season, reaching up to 45 °C, can pose challenges to the use of LLINs, as people may prefer to sleep outside the huts to escape the heat. It is important to consider the local context and climate conditions when implementing vector control interventions. In regions with extreme heat, alternative strategies may be needed to ensure effective protection against malaria.

The use of other mosquito-biting control tools, such as mosquito coils, domestic insecticide sprays, and traditional smoking methods, is high in Djougou (urban area) where LLINs use is relatively low. This suggests that these alternative tools were used as substitutes or in combination with LLINs. In the urban area (Djougou), where LLINs use was poorer (39% of the population), the reliance on alternative tools was higher (83% of the population). Several reasons may support this observation. One possibility is that some individuals may perceive the alternative tools to be more effective or convenient in certain situations, such as during hot weather or when sleeping outside the huts. They may choose to use these tools instead of or in addition to LLINs in order to enhance their personal protection against mosquito bites. Additionally, cultural practices and beliefs, as well as the availability and accessibility of different tools, may influence the choices of individuals in using mosquito-biting control measures, especially in the urban area. Factors such as cost, ease of use, and personal preferences can also play a role in determining the selection of specific tools against vector biting [34]. However, it is crucial to ensure that these alternative tools are used in a safe manner [35,36]. The contrasting usage patterns of other mosquito bite control tools between the urban and rural areas can be attributed to various factors, including socio-economic differences, access to alternative tools, and the implementation of IRS. In the rural area, where LLINs use was combined with IRS, the reliance on other tools such as mosquito coils, domestic insecticide sprays, and traditional smoking methods was relatively low (23% of the population). This could be attributed to the successful implementation of IRS, which involves the application of insecticides to indoor surfaces to kill mosquitoes. The presence of IRS in the rural area may have significantly reduced the mosquito population, leading to a decreased need for additional mosquito-biting control tools, as some of them may have limitations in providing the expected protection. Further research and interventions could focus on understanding the factors influencing the preference for alternative tools over LLINs and finding ways to address barriers to LLINs use in the specific context of high temperatures during the dry season or when IRS was applied.

### 4.2. Limited Protection of Long-Lasting Insecticide-Treated Nets and Indoor Residual Spaying

Benin remains a malaria-endemic country and has faced challenges in reducing malaria cases and deaths in recent years between 2015 and 2021 [2]. Globally, the use of LLINs alone has not provided protection against malaria. In 2021, a total of 2,656,855 cases of malaria were recorded, accounting for 43.5% of consultations and hospitalisations [37]. The fact that malaria remains the main reason for seeking care and hospitalisation in Benin, regardless of age group, highlights the ongoing burden of malaria in the country. This indicates that despite efforts to control and prevent malaria, it continues to have a significant impact on the health system and population. Vector control is more than just spraying insecticides or delivering nets [38]. Before scaling up, vector control interventions are not adapted to the country’s profile, and not, therefore, to the epidemiological facies and the intrinsic effectiveness of insecticides to wild local vectors; the living environment, particularly the type of housing; and the behaviour of the population. The intrinsic effectiveness of insecticides against local mosquito species is crucial for achieving successful vector control. Different regions may have varying vector species and their susceptibility to insecticides, so it is important to consider these factors when validating, selecting, and deploying insecticide-based interventions. The living environment, including the type of housing, can also impact the effectiveness of vector control interventions. Housing characteristics such as the presence of screens, open eaves, or proper ventilation can influence mosquito entry and the effectiveness of measures like LLINs or IRS [33]. Additionally, understanding the behaviour and practices of the population is essential for the successful implementation of vector control interventions. Knowledge of local habits, sleeping patterns, and outdoor activities can help optimise the use of mosquito nets and inform targeted interventions. Also, the effectiveness of vector control interventions can be hindered by weak epidemiological surveillance systems. Insufficient financial and logistical resources can limit the comprehensive monitoring of disease prevalence, incidence, intervention coverage, and actual use of mosquito nets. Robust surveillance systems need to be established to collect reliable data and inform evidence-based decision making. Also, the monitoring of the sensitivity of insecticides and detecting changes in parasite genetic profiles and resistance mechanisms are crucial aspects of surveillance and monitoring. Regular assessment of insecticide efficacy and monitoring of parasite resistance can guide the selection and deployment of appropriate interventions. In summary, adapting vector control interventions to the country’s profile, considering local vector species, housing characteristics, population behaviour, and strengthening epidemiological surveillance, are important steps to enhance the effectiveness of malaria control efforts. 

Up to 90% of urban population growth occurs in Asia and Africa [14]. As malaria elimination remains a major challenge, understanding the epidemiological, environmental, and socioeconomic determinants of malaria will be essential to improving the response to malaria in urban areas. Social determinants can include the acceptability of interventions, place of residence, household wealth, poverty, gender, and education [9]. Djougou city is characterised by a moderate malaria burden, moderate or low acceptability of LLINs, poor quality housing, and polluted/stagnant pools. High heterogeneity of LLINs protection was observed from 0% to 85%. Place of residence can determine the protection conferred by LLINs in the urban setting because in the same district, the LLINs allow for protection in some neighbourhoods instead of others. As in Benin, malaria prevalence varied across very small geographic distances [39] in the Democratic Republic of the Congo. If the village/neighbourhood level interacts highly between LLINs use and the malaria burden, then it makes more sense to consider the malaria heterogeneity index (hot/cold spots) at the village/neighbourhood level (the smallest demographic unit possible) in the algorithm of the selection of interventions [40]. Beyond micro-stratification [20], a more precise response can be provided to decision makers and indicates precisely which control tool is suitable or not effective at the micro level. Even if this approach requires better organisation and additional costs, it would contribute to LLINs effectiveness and be a decisive step towards malaria elimination. 

Cobly, a rural area, is characterised by flooded grasslands and large pools. The area is a high-transmission setting with moderate use of LLINs. Most houses have open eaves that allow high levels of indoor mosquito biting. The presence of open eaves results in higher malaria vector entry into houses, infection, and cases [33,41,42]. LLINs only seem not to be sufficient in this context. Housing improvements as malaria control interventions can be implemented by updating laws on housing and promoting community engagement. For example, an appropriate housing design without eaves can be developed and awareness programmes can be provided for household residents that, for example, encourage people to close their doors early in the evenings [33]. The use of LLINs alone, IRS implementation alone, or IRS + LLINs did not give additional protection in the rural area compared to no intervention. This is worrying as LLINs remain the most implemented malaria control tool in still-endemic countries. Poor use or poor condition of the net [43] or resistance of the malaria vectors to insecticides [44] are often cited as reasons for poor or ineffective LLINs and IRS. The quality of application of insecticides and effective remanence of the product should be questioned regularly. Compared to LLINs alone, IRS alone and LLINs + IRS seemed more efficient but IRS has limitations concerning cost and budgets. Logistics, domestic contribution, and monitoring are also difficult to manage [45]. The local applicators of insecticides frequently divert products for selling to private use in fields for growing food and other industrial crops. Recent studies showed that, compared with LLINs, IRS was about five times more expensive per person protected per year. This means that IRS is considerably less cost-effective than expected [2,45,46]. Factors related to vector behaviour were not explicitly considered in this study but were clearly highlighted in Benin [11]. The multi-resistance of *Anopheles* spp. to the range of insecticides used for bed-nets and IRS [47] has been widely studied. However, strong resistance to pyrethroid insecticides persists without convincingly demonstrating their role in reducing their protection effect against malaria infection and disease by using randomised trials [48]. Recently, under the best conditions of utilisation, the effect of the latest generation of LLINs, synergist nets combining pyrethroid (Py) and piperonyl-butoxide (PBO), was not greater than 50% of protection [49].

Innovative and visionary approaches to malaria control other than those using insecticides on the front line should be promoted and deployed in the sub-Saharan region. Vector control adapted to local conditions should now be the most reasonable approach to sustainable malaria control. For example, malaria case management remains an issue in many countries and must become the cornerstone of malaria elimination.

Modelling the impact of all possible control measures combinations (LLINs, IRS, chemoprevention, case management, awareness, etc.) and their benefits according to local area characteristics could be one of the future research directions. The development of mathematical models to predict the effects and impact of malaria control interventions can be a valuable tool for decision making and resource allocation. These models allow researchers and policymakers to assess the potential outcomes of different intervention strategies, estimate their cost-effectiveness, and prioritise resource allocation based on the predicted impact. The development of mathematical models should not be seen as a waste of funds; rather, investment in the development and refinement of such models can yield significant benefits in terms of informed decision making, optimising resource allocation, and maximising the impact of interventions. Mathematical models provide cost-effective ways to explore different scenarios and evaluate the potential effectiveness of interventions before their implementation. They can help identify strategies that are most likely to achieve desired outcomes, reduce the risk of investing in ineffective or suboptimal interventions, and guide the allocation of limited resources to areas where they can have the greatest impact. The development and utilisation of mathematical models should be accompanied by robust data collection, validation, and continuous refinement to ensure their accuracy and relevance. Models should be regularly updated with new local data and adapted to reflect the changing dynamics of malaria transmission and intervention strategies. 

### 4.3. Focus on Neglected Social Groups: Older Children, Adolescents, and Males

According to the findings, in both rural and urban communities, age and gender determine the malaria burden. Age groups 5–9 years and 10–14 years were associated with a high burden of malaria infection and UCC compared to the youngest (under 5 years old). When malaria transmission declines mostly in moderate transmission areas, the age pattern generally extends or shifts, and the malaria burden increases among older children, adolescents, and young adults; even severe clinical cases can occur in the youngest [9,12,13,50,51]. Similarly, in another region of Benin (centre region), a recent population-based study found that malaria infection was higher in adolescents than children aged under 5 years [52]. Thus, malaria control policies should be improved by enlarging both behaviour change communication strategies and providing free access to UCC management for the 5–14 years age group. The study also showed that women had lower rates of infection and suffered less from malaria cases than men. A possible reason for this could be differences in the lifestyle habits of both genders. A longitudinal study conducted in the peri-urban area of Cotonou in southern Benin showed that females had a lower risk of UCC than males [10]. Women are naturally more sensitive to health issues because they are prior targets of communication and more involved in the implementation of interventions [53]. It is, therefore, time to turn attention to the burden of malaria among adolescents and its consequences for their health condition and to review the global malaria programme elimination strategy with more consideration towards older children, adolescents, and men concerning intervention, use of interventions, and benefits.

## 5. Limitations

This study was carried out in 2014. According to the latest World Malaria Report 2022, the incidence of malaria has remained stagnant in Benin. IRS intervention has stopped in the last 3 years, and LLINs impregnated with pyrethroids will no be longer distributed. Other issues such as checking the intrinsic effectiveness of insecticides against local mosquito species and setting up a robust surveillance system are yet to be addressed and resolved. These data are, therefore, useful for understanding the various factors that could explain the persistence of a high burden of malaria at the country level and future challenges to eliminating malaria in Benin. Currently, there are no recent data at the field scale in Benin in general, especially in the northern region where the burden of malaria is the greatest, to achieve this analysis.

The information bias inherent in questionnaire surveys has not been completely eliminated. We assumed that people may overestimate the number of times they have slept under an insecticide-treated net as a result of being asked a question in a survey.

## 6. Conclusions

The use of LLINs combined or not with IRS implementation did not provide significant protection against malaria infection and UCC when compared to no intervention. IRS alone or in combination with LLINs showed greater effectiveness in reducing malaria compared to LLINs alone. However, the implementation of IRS at a large scale presents challenges. These results support the argument that we need to be looking beyond LLINs for malaria control by continuing innovation in malaria vector control tool design. This study also revealed that older children, adolescents, and men were more susceptible to malaria compared to younger children and women, respectively. This highlights the importance of considering age and gender-specific factors in malaria prevention and control efforts. Furthermore, the study identified the significance of place of residence, house design, and socio-economic barriers in urban areas as contributing factors to the malaria burden. These findings suggest that addressing environmental and housing-related factors, as well as addressing socio-economic challenges, can play a crucial role in malaria elimination efforts. They further suggest that mathematical models can be developed to predict the effects of malaria control interventions as a valuable investment that can help optimise resource allocation, inform decision making, and maximise the impact of interventions.

## Figures and Tables

**Table 1 tropicalmed-08-00475-t001:** (**a**) Socio-demographic and economic characteristics of inhabitants and households, Djougou, urban area, 2014. (**b**) Socio-demographic and economic characteristics of inhabitants and households, Cobly, rural area, 2014.

(a)			
**Variables**	**Djougou (Urban Area)**		
	**n = 2080**	**%**	**CI95%**
**Individual-level**			
Age (years)			
0–4	480	23.1	21.3–25.0
5–9	441	21.2	19.4–23.0
10–14	314	15.1	13.6–16.8
≥15	845	40.6	38.5–42.7
Gender			
Male	921	44.3	42.2–46.5
Female	1159	55.7	53.5–57.8
**Household-level**			
Religion of the head of household			
None	8	0.4	0.2–0.8
Traditional	35	1.7	1.2–2.4
Muslim	1877	90.2	88.8–91.4
Christian	160	7.7	6.6–8.9
Educational status of the head of household			
Illiterate	1292	62.1	50.0–64.2
Primary	391	18.8	17.1–20.6
Secondary (1st level)	193	9.3	8.1–10.7
Secondary (2nd level)	146	7.0	5.9–8.1
University	58	2.8	2.2–3.7
Wealth index			
Most poor	483	23.2	21.4–25.0
Very poor	416	20.0	18.3–21.8
Poor	416	20.0	18.3–21.8
Less poor	349	16.8	15.3–18.5
Least poor	416	20.0	18.3–21.8
Educational status of the respondent			
Illiterate	1708	82.1	80.4–83.7
Primary	150	7.2	6.1–8.4
Secondary (1st level)	127	6.1	5.1–7.2
Secondary (2nd level)	81	3.9	3.2–4.9
University	14	0.7	0.4–1.2
Household size (inhabitants)			
1–5	616	29.6	27.6–31.6
6–8	782	37.6	35.6–39.8
≥9	682	32.8	30.8–34.9
Window with screen or net			
No	1613	77.5	75.7–79.3
Yes	467	22.5	20.7–24.3
Windows and doors closed at night			
No	830	39.9	37.8–42.1
Yes	1250	60.1	57.9–62.2
Open eaves			
No	1877	90.2	88.9–91.5
Yes	203	9.8	8.5–11.1
(**b**)			
**Variables**	**Cobly (Rural Area)**		
	**n = 2770**	**%**	**CI95%**
**Individual-level**			
Age (years)			
*0–4*	540	19.5	18.0–21.0
*5–9*	564	20.3	18.9–21.9
*10–14*	392	14.1	12.9–15.5
*15–40*	932	33.7	31.9–35.5
*≥41*	342	12.4	11.2–13.7
Gender			
*Male*	1384	49.9	48.0–51.8
*Female*	1386	50.1	48.1–51.9
**Household-level**			
Religion of the head of household			
*None*	335	12.1	10.9–13.4
*Traditional*	1366	49.3	47.4–51.2
*Muslim*	83	3.0	2.4–3.7
*Christian*	986	35.6	33.8–37.4
Educational status of the head of household			
*Illiterate*	2104	76.0	74.3–77.5
*Primary*	361	13.0	11.8–14.4
*Secondary (1st level)*	217	7.8	6.9–8.9
*Secondary (2nd level)*	71	2.6	2.0–3.2
*University*	17	0.6	0.4–1.0
Wealth index			
*Most poor*	585	21.1	19.6–22.7
*Very poor*	581	21.0	19.5–22.6
*Poor*	544	19.6	18.2–21.2
*Less poor*	528	19.1	17.6–20.6
*Least poor*	532	19.2	17.8–20.7
Educational status of the respondent			
*Illiterate*	2348	84.8	83.1–85.8
*Primary*	220	7.9	6.9–9.0
*Secondary (1st level)*	150	5.7	4.9–6.6
*Secondary (2nd level)*	43	1.6	1.1–2.1
*University*	9	0.3	0.2–0.6
Household size (inhabitants)			
*1–5*	561	20.3	18.9–21.8
*6–8*	1211	43.7	41.9–45.6
*≥9*	998	36.0	34.2–37.9
Window with screen or net			
*No*	2736	98.8	98.3–99.1
*Yes*	34	1.2	0.9–1.7
Windows and doors closed at night			
*No*	378	13.6	12.4–20.3
*Yes*	2392	86.4	85.0–87.6
Open eaves			
*No*	1938	70.0	68.2–71.7
*Yes*	832	30.0	28.3–31.8

**Table 2 tropicalmed-08-00475-t002:** (**a**) Long-lasting insecticide-treated nets use associated with malaria infection, univariate and multivariate analyses, Djougou, Benin, 2014. (**b**) Long-lasting insecticide-treated nets use associated with malaria clinical uncomplicated cases, univariate and multivariate analyses, Djougou, Benin, 2014.

(a)								
	**Infection Rate (Djougou Commune)**							
**Factors**					**Univariate**		**Multivariate**	
	**Case**		**Control**		**Crude OR [CI95%]**	** *p* **	**Adjusted OR [CI95%]**	** *p* **
	**n**	**%**	**n**	**%**				
**Individual-level**								
Age (years)						<0.0001		<0.0001
0–4	120	25.59	349	74.41	1		1	
5–9	164	37.61	272	62.39	1.64 [1.24–2.17]	0.0005	1.70 [1.27–2.28]	0.0004
10–14	138	44.09	175	55.91	2.16 [1.60–2.92]	<0.0001	2.34 [1.70–3.22]	<0.0001
15–40	133	20.43	518	79.57	0.70 [0.53–0.92]	0.0134	0.77 [0.57–1.03]	0.088
≥41	28	14.51	165	85.49	0.46 [0.29–0.72]	0.0008	0.49 [0.31–0.79]	0.0031
Gender								
Male	294	32.06	623	67.94	1		1	
Female	297	25.69	859	74.31	0.73 [0.60–0.88]	0.0014	0.77 [0.63–0.95]	0.018
LLINs use the previous week								
No (sometimes or never)	337	29.36	811	70.64	1		1	
Yes (all nights)	254	27.46	671	72.54	0.91 [0.75–1.10]	0.3418	1.80 [1.01–3.21]	0.0429
Sleeping outdoor								
No	582	28.57	1455	71.43	1			
Yes	9	25.00	27	75.00	0.83 [0.38–1.78]	0.8661		
Use of anti-malarial drug the 2 previous weeks								
No	455	27.91	1175	72.09	1			
Yes	136	30.70	307	69.30	1.14 [0.90–1.43]	0.2495		
**Household-level**								
Educational status of the respondent						0.0299		
Illiterate	399	31.03	887	68.97	1			
Primary	96	24.55	295	75.49	0.72 [0.55–0.93]	0.0142		
Secondary (1st level)	47	24.35	146	75.65	0.71 [0.50–1.01]	0.0605		
Secondary (2nd level)	36	25.00	108	75.00	0.74 [0.49–1.10]	0.1372		
University	13	22.03	46	77.97	0.62 [0.33–1.17]	0.1463		
Household size (inhabitants)						0.0257		
1–5	34	22.08	120	77.92	1			
6–8	282	27.14	757	72.86	1.31 [0.87–1.97]	0.1852		
≥9	275	31.25	605	68.75	1.60 [1.06–2.40]	0.0227		
Wealth index						<0.0001		<0.0001
Most poor	158	38.07	257	61.93	1		1	
Very poor	127	30.53	289	69.47	0.71 [0.53–0.95]	0.0222	0.69 [0.51–0.94]	0.0194
Poor	119	29.75	281	70.25	0.68 [0.51–0.92]	0.0123	0.64 [0.46–0.88]	0.007
Less poor	104	23.74	334	76.26	0.50 [0.37–0.68]	<0.0001	0.48 [0.34–0.68]	<0.0001
Least poor	83	20.54	321	79.46	0.42 [0.30–0.57]	<0.0001	0.45 [0.31–0.65]	<0.0001
Use of other tools against vector biting								
No	65	22.57	223	77.43	1			
Yes	526	29.47	1259	70.53	1.43 [1.06–1.92]	0.0161		
Screened windows								
No	458	28.53	1147	71.47	1			
Yes	133	28.42	335	71.58	0.99 [0.79–1.24]	0.9606		
Open eaves								
No	153	10.32	1329	89.88	1			
Yes	49	8.29	542	91.71	0.78 [0.56–1.09]	0.1588		
Place of residence						<0.0001		<0.0001
Angaradébou					1		1	
Baparappé					0.77 [0.52–1.13]	0.1885	0.78 [0.44–1.38]	0.400
Killir					0.57 [0.38–0.86]	0.0078	0.72 [0.39–1.32]	0.2932
Leman mandé					0.57 [0.39–0.84]	0.0049	1.12 [0.67–1.86]	0.6461
Sassirou					0.62 [0.42–0.93]	0.0229	1.05 [0.52–2.12]	0.8743
Tim tim Bongo					0.23 [0.14–0.36]	<0.0001	0.34 [0.19–0.61]	0.0003
Zembougou Béri					0.38 [0.25–0.56]	<0.0001	0.55 [0.33–0.91]	0.0207
Zongo					0.40 [0.26–0.60]	<0.0001	0.74 [0.42–1.29]	0.2917
Zountori					0.45 [0.30–0.67]	0.0001	0.72 [0.39–1.32]	0.2959
LLINs use the previous week * (Angaradébou/Baparappé)							0.88 [0.38–1.99]	0.7623
LLINs use the previous week * (Angaradébou/Killir)							0.47 [0.20–1.10]	0.0848
LLINs use the previous week * (Angaradébou/Leman mandé)							0.15 [0.06–0.38]	0.0001
LLINs use the previous week * (Angaradébou/Sassirou)							0.35 [0.14–0.86]	0.0228
LLINs use the previous week * (Angaradébou/Tim tim Bongo)							0.68 [0.23–2.01]	0.4900
LLINs use the previous week * (Angaradébou/Zembougou Béri)							0.35 [0.13–0.89]	0.0275
LLINs use the previous week * (Angaradébou/Zongo)							0.45 [0.19–1.05]	0.0666
LLINs use the previous week * (Angaradébou/Zountori)							0.32 [0.13–0.75]	0.0089
(**b**)								
	**Uncomplicated Clinical Cases (Djougou Commune)**							
**Factors**					**Univariate**		**Multivariate**	
	**Case**		**Control**		**Crude OR [CI95%]**	** *p* **	**Adjusted OR [CI95%]**	** *p* **
	**n**	**%**	**n**	**%**				
**Individual-level**								
Age (years)						0.0001		0.0042
0–4	40	8.35	439	91.65	1		1	
5–9	52	11.90	385	88.10	1.48 [0.96–2.28]	0.0757	1.50 [0.96–2.35]	0.0714
10–14	40	12.78	273	87.22	1.60 [1.01–2.55]	0.0446	1.74 [1.07–2.81]	0.0239
15–40	37	5.68	614	94.32	0.66 [0.41–1.05]	0.0804	0.70 [0.43–1.13]	0.1477
≥41	9	4.66	184	95.34	0.53 [0.25–1.12]	0.1009	0.55 [0.26–1.18]	0.1267
Gender								
Male	94	10.25	823	89.75	1			
Female	84	7.27	1072	92.73	0.68 [0.50–0.93]	0.0160		
LLINs use the previous week								
No (sometimes or never)	99	8.62	1049	91.38	1		1	
Yes (all nights)	79	8.54	846	91.46	0.98 [0.72–1.34]	0.9464	0.95 [0.67–1.35]	0.8154
Sleeping outdoor								
No	174	8.54	1863	91.46	1			
Yes	4	11.11	32	88.89	1.33 [0.46–3.82]	0.3739		
Use of anti-malarial drug the 2 previous weeks								
No	156	8.85	1606	91.15	1			
Yes	22	7.07	289	92.93	0.78 [0.49–1.24]	0.3017		
**Household-level**								
Educational status of the respondent								
Illiterate	129	10.03	1157	89.97	1			
Primary	23	5.88	368	94.12	0.56 [0.35–0.88]	0.0134		
Secondary (1st level)	12	6.22	181	93.78	0.59 [0.32–1.09]	0.0959		
Secondary (2nd level)	8	5.56	136	94.44	0.52 [0.25–1.10]	0.0886		
University	6	10.17	53	89.83	1.01 [0.42–2.40]	0.9724		
Household size (inhabitants)								
1–5	8	5.19	146	94.81	1			
6–8	91	8.76	948	91.24	1.75 [0.83–3.68]	0.1394		
≥9	79	8.98	801	91.02	1.79 [0.85–3.80]	0.1237		
Wealth index								
Most poor	52	12.53	363	87.47	1		1	
Very poor	43	10.34	373	89.66	0.80 [0.52–1.23]	0.3211	0.79 [0.50–1.24]	0.3131
Poor	32	8.00	368	92.00	0.60 [0.38–0.96]	0.0348	0.62 [0.38–1.02]	0.0627
Less poor	23	5.25	415	94.75	0.38 [0.23–0.64]	0.0003	0.38 [0.22–0.67]	0.0007
Least poor	28	6.93	376	93.07	0.51 [0.32–0.84]	0.0077	0.57 [0.33–0.98]	0.0458
Use of other tools against vector biting								
No	26	9.03	262	90.97	1			
Yes	152	8.52	1633	91.48	0.93 [0.60–1.45]	0.8613		
Screened windows								
No	136	8.47	1469	91.53	1			
Yes	42	8.97	426	91.03	1.06 [0.74–1.53]	0.7336		
Open eaves								
No	154	8.23	1717	91.77	1		1	
Yes	24	11.88	178	88.12	1.50 [0.95–2.37]	0.0785	1.70 [1.02–2.82]	0.0405
Place of residence								<0.0001
Angaradébou	17	7.98	196	92.02	1		1	
Baparappé	21	9.59	198	90.41	1.22 [0.62–2.38]	0.5558	1.31 [0.66–2.59]	0.4316
Killir	16	7.88	187	92.12	0.98 [0.48–2.00]	0.9701	0.88 [0.42–1.84]	0.7473
Leman mandé	28	11.24	221	88.76	1.46 [0.77–2.74]	0.2403	1.52 [0.78–2.96]	0.2160
Sassirou	35	17.16	169	82.84	2.38 [1.29–4.41]	0.0055	0.92 [0.43–1.96]	0.0062
Tim tim Bongo	15	6.41	219	93.56	0.78 [0.38–1.62]	0.5207	0.92 [0.43–1.96]	0.8365
Zembougou Béri	28	10.33	243	89.67	1.32 [0.70–2.49]	0.3779	1.34 [0.70–2.58]	0.3719
Zongo	9	3.70	234	96.30	0.44 [0.19–1.01]	0.0549	0.62 [0.26–1.49]	0.2922
Zountori	9	3.80	228	96.20	0.45 [0.19–1.04]	0.0632	0.42 [0.18–0.98]	0.0463

LLINs: long-lasting insecticide-treated nets; * Interaction between malaria infection rate and LLINs use according to the place of residence.

**Table 3 tropicalmed-08-00475-t003:** (**a**) LLINs use combined or not with IRS associated with malaria infection, univariate and multivariate analyses, Cobly, Benin, 2014. (**b**) LLINs combined with IRS associated with malaria uncomplicated clinical cases, univariate and multivariate analyses, Cobly, Benin, 2014.

(a)								
	**Infection Rate (Cobly Commune)**							
**Factors**					**Univariate**		**Multivariate**	
	**Case**		**Control**		**Crude OR [CI95%]**	** *p* **	**Adjusted OR [CI95%]**	** *p* **
	**n**	**%**	**n**	**%**				
**Individual-level**								
Age (years)						<0.0001		<0.0001
*0–4*	240	49.08	249	50.92	1		1	
5–9	394	77.25	116	22.75	3.52 [2.68–4.62]	<0.0001	3.62 [2.73–4.79]	<0.0001
10–14	233	74.68	79	25.32	3.06 [2.24–4.17]	<0.0001	3.08 [2.23–4.25]	<0.0001
15–40	222	35.58	402	64.42	0.57 [0.45–0.72]	<0.0001	0.60 [0.47–0.78]	0.0001
≥41	51	21.43	187	78.57	0.28 [0.19–0.40]	<0.0001	0.27 [0.19–0.39]	<0.0001
Gender								
Male	591	57.49	437	42.51	1		1	
Female	546	47.81	596	52.19	0.67 [0.57–0.80]	<0.0001	0.79 [0.65–0.96]	0.0177
**Vector control measures use**						0.0406		0.2030
No LLINs or IRS	38	53.52	33	46.48	1		1	
LLINs alone	43	60.56	28	39.44	1.33 [0.68–2.59]	0.3971	1.01 [0.47–2.15]	0.9968
IRS alone	480	55.36	387	44.64	1.07 [0.66–1.74]	0.7641	1.03 [0.58–1.81]	0.9106
LLINs combined with IRS	579	49.74	585	50.26	0.85 [0.53–1.39]	0.5367	0.75 [0.42–1.31]	0.3158
Sleeping outdoor								
No	1109	53.04	982	46.96	1			
Yes	31	37.80	51	62.20	0.53 [0.34–0.84]	0.0067		
Use of anti-malarial drug the 2 previous weeks								
No	1057	52.88	942	47.12	1			
Yes	83	47.70	91	52.30	0.81 [0.59–1.10]	0.1898		
**Household-level**								
Educational status of the respondent						<0.0001		0.0038
Illiterate	896	55.27	725	44.73	1		1	
Primary	137	47.74	150	52.26	0.73 [0.57–0.95]	0.0184	0.80 [0.60–1.07]	0.1444
Secondary school (1st level)	87	46.03	102	53.97	0.69 [0.51–0.93]	0.0162	0.77 [0.54–1.09]	0.1545
Secondary school (2nd level)	16	26.67	44	73.33	0.29 [0.16–0.52]	<0.0001	0.39 [0.21–0.75]	0.0047
University	4	25.00	12	75.00	0.26 [0.08–0.83]	0.0237	0.32 [0.09–1.13]	0.0774
Household size (inhabitants)								
1–5	45	42.45	61	57.55	1			
6–8	506	50.85	489	49.15	1.40 [0.93–2.10]	0.1013		
≥9	589	54.94	483	45.06	1.65 [1.10–2.47]	0.0146		
Wealth index								
Most poor	225	53.57	195	46.43	1			
Very poor	248	56.24	193	43.76	1.11 [0.85–1.45]	0.4323		
Poor	232	53.88	199	46.17	1.01 [0.77–1.32]	0.9401		
Less poor	226	52.19	207	47.81	0.94 [0.72–1.23]	0.687		
Least poor	209	46.65	239	53.35	0.75 [0.58–0.98]	0.0418		
Use of other tools against vector biting								
No	1066	53.03	944	46.97	1			
Yes	74	45.40	89	54.60	0.73 [0.53–1.01]	0.0604		
Screened windows								
No	16	1.55	1017	98.45	1			
Yes	14	1.23	1126	98.77	0.79 [0.38–1.62]	0.5221		
Open eaves								
No	321	31.07	712	68.93	1			
Yes	335	29.39	805	70.61	0.92 [0.76–1.10]	0.3918		
Place of residence								<0.0001
Koukontouga	184	73.60	66	26.40	1		1	
Nouangou	145	47.54	160	52.46	0.32 [0.22–0.46]	<0.0001	0.33 [0.22–0.50]	<0.0001
Ouorou	134	48.38	143	51.62	0.33 [0.23–0.48]	<0.0001	0.35 [0.23–0.52]	<0.0001
Pintinga	92	62.59	55	37.41	0.60 [0.38–0.92]	0.0219	0.70 [0.43–1.14]	0.1555
Tapoga	96	42.29	131	57.71	0.26 [0.17–0.38]	<0.0001	0.28 [0.18–0.43]	<0.0001
Touga	137	52.69	123	47.31	0.39 [0.27–0.57]	<0.0001	0.46 [0.30–0.69]	<0.0001
Yimpisséri I	126	56.00	99	44.00	0.45 [0.31–0.67]	0.0001	0.46 [0.30–0.69]	0.0001
Ympisséri II	97	50.79	94	49.21	0.37 [0.24–0.55]	<0.0001	0.41 [0.26–0.63]	<0.0001
Zanouiri	129	44.33	162	55.67	0.28 [0.19–0.41]	<0.0001	0.27 [0.18–0.40]	<0.0001
(**b**)								
	**Uncomplicated Clinical Cases (Cobly Commune)**							
**Factors**					**Univariate**		**Multivariate**	
	**Case**		**Control**		**Crude OR [CI95%]**	** *p* **	**Adjusted OR [CI95%]**	** *p* **
	**n**	**%**	**n**	**%**				
**Individual-level**								
Age (years)						<0.0001		<0.0001
0–4	61	12.47	428	87.53	1		1	
5–9	104	20.39	406	79.61	1.79 [1.27–2.53]	0.0008	1.93 [1.35–2.74]	0.0002
10–14	54	17.31	258	82.69	1.46 [0.98–2.18]	0.0581	1.51 [1.00–2.27]	0.0466
15–40	43	6.89	581	93.11	0.51 [0.34–0.78]	0.0017	0.52 [0.34–0.80]	0.0028
≥41	5	2.10	233	97.90	0.15 [0.05–0.38]	0.0001	0.16 [0.06–0.41]	0.0001
Gender								
Male	142	13.81	886	86.19	1			
Female	123	10.77	1019	89.23	0.75 [0.58–0.97]	0.0306		
**Vector control measures use**						0.0052		0.1000
No LLINs or IRS	11	15.49	60	84.51	1		1	
LLINs alone	18	25.35	53	74.65	1.85 [0.80–4.27]	0.1483	1.27 [0.52–3.10]	0.5928
IRS alone	98	11.30	769	88.70	0.69 [0.35–1.38]	0.2919	0.61 [0.30–1.26]	0.1909
LLINs combined with IRS	140	12.03	1024	87.97	0.74 [0.38–1.45]	0.3884	0.59 [0.28–1.20]	0.1480
Sleeping outdoor								
No	261	12.48	1830	87.52	1			
Yes	6	7.32	76	92.68	0.55 [0.23–1.28]	0.1622		
Use of anti-malarial drug the 2 previous weeks								
No	250	12.51	1749	87.49	1			
Yes	17	9.77	157	90.23	0.75 [0.45–1.27]	0.1518		
**Household-level**								
Educational status of the respondent						0.0014		0.0043
Illiterate	210	12.95	1411	87.05	1		1	
Primary	22	7.67	265	92.33	0.55 [0.35–0.88]	0.0126	0.53 [0.33–0.86]	0.0103
Secondary school (1st level)	33	17.46	156	82.54	1.42 [0.95–2.12]	0.0869	1.45 [0.93–2.26]	0.0986
Secondary school (2nd level)	2	3.33	58	96.67	0.23 [0.05–0.95]	0.0431	0.24 [0.05–1.05]	0.0534
University	0	0.00	16	100.00	-	0.9673	-	0.9644
Household size (inhabitants)								
1–5	8	7.55	98	92.45	1			
6–8	124	12.46	871	87.54	1.74 [0.82–3.67]	0.1433		
≥9	135	12.59	937	87.41	1.76 [0.83–3.71]	0.1339		
Wealth index								
Most poor	58	13.81	362	86.19	1			
Very poor	57	12.93	384	87.07	0.92 [0.62–1.37]	0.7030		
Poor	55	12.76	376	87.24	0.91 [0.61–1.35]	0.6500		
Less poor	51	11.78	382	88.22	0.83 [0.55–1.24]	0.3700		
Least poor	46	10.27	402	89.73	0.71 [0.47–1.07]	0.1095		
Use of other tools against vector biting								
No	243	12.09	1767	87.91	1			
Yes	24	14.72	139	85.28	1.25 [0.79–1.97]	0.3245		
Screened windows								
No	265	12.37	1878	87.63	1			
Yes	2	6.67	28	93.33	0.50 [0.11–2.13]	0.3450		
Open eaves								
No	169	11.14	1348	88.86	1			
Yes	98	14.94	558	85.06	1.40 [1.07–1.83]	0.0132	1.59 [1.19–2.13]	0.0017
Place of residence						0.0104		0.0112
Koukontouga	47	18.80	203	81.20	1			
Nouangou	37	12.13	268	87.87	0.59 [0.37–0.95]	0.0303	0.65 [0.39–1.08]	0.0981
Ouorou	26	9.39	251	90.61	0.44 [0.26–0.74]	0.0021	0.54 [0.31–0.94]	0.0299
Pintinga	21	14.29	126	85.71	0.71 [0.41–1.26]	0.2503	1.01 [0.55–1.82]	0.9733
Tapoga	29	12.78	198	87.22	0.63 [0.38–1.04	0.0741	0.78 [0.45–1.34]	0.3737
Touga	39	15.00	221	85.00	0.76 [0.47–1.21]	0.2527	0.97 [0.59–1.59]	0.9077
Yimpisséri I	23	10.22	202	89.78	0.49 [0.28–0.84]	0.0094	0.49 [0.28–0.87]	0.0158
Ympisséri II	21	10.99	170	89.01	0.53 [0.30–0.92]	0.0261	0.66 [0.37–1.18]	0.1697
Zanouiri	24	8.25	267	91.75	0.38 [0.22–0.65]	0.0004	0.46 [0.27–0.80]	0.0066

LLINs: long-lasting insecticide-treated nets; IRS: indoor residual spraying, UCC: uncomplicated clinical cases.

## Data Availability

All relevant data supporting the conclusion of this article can be provided as an additional file.

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
