# Peer review of "Long-Lasting Insecticide-Treated Nets Combined or Not with Indoor Residual Spraying May Not Be Sufficient to Eliminate Malaria: A Case-Control Study, Benin, West Africa"

_tropicalmed, 2023, doi:10.3390/tropicalmed8100475_

Round 1
Reviewer 1 Report (New Reviewer)
The manuscript “Long-lasting insecticide-treated nets combined or not with indoor residual spraying, may not be sufficient to eliminate malaria: a case-control study, Benin, West Africa” describes a rigorous case-control study that, unfortunately, demonstrates very little to no effect of long-lasting insecticide-treated nets (LLIN), indoor residual spraying (IRS) or a combination of the two, on either the risk of malaria infection or the risk of uncomplicated cases (UCC). These studies are important because it is critical to continually evaluate our control methods, either as a trigger to improve them or as justification to focus our resources in other areas.
In general, the study is well-designed and addresses an extremely important public health issue. Most of my questions concern either the description of the methodology or the interpretation of the results. Specific comments are below:
INTRODUCTION
1. Lines 81-82. There is a duplicated sentence here.
MATERIALS AND METHODS
2. Lines 99-118 This first paragraph covers a number of different topics. I suggest breaking it down into multiple paragraphs.
3. Line 101. 48.97% is in two significant digits. The other numbers presented in this section are only taken out to one significant digit. I suggest using a consist format for decimals.
4. Line 126. What does NMCP stand for?
5. Lines 138-140. It seems like there is some repetition here (e.g. “to carry out large scale operations”). See if you can find a way to streamline this.
6. Lines 156-161. How many times was a random point on a map chosen per village/neighborhood? Did you choose a random area and just keep visiting second households on the right until all 50 houses were chosen? Or did you get to the end of the street or have to stop visiting for some other reason before getting to 50, and then choosing an additional random point on the map?
7. Lines 182-184 In the survey, was anything done to address the possibility that people might overestimate the number of times they slept under a LLIN as a result of being asked in a survey?
8. Lines 204-227 The detailed description of how the diagnostic tests work is not necessary for a study like this. The procedure is well-established, so a brief description and citation to previous work is sufficient.
9. Lines 232-236. This is more a question for Results, but how many severe clinical cases were observed in the study?
10. Line 239. I understand the definition of co-variables to mean continuous variables. Age group, gender and place of residence are discrete variables. Did you mean to say co-variables (or do I have the definition wrong)?
DISCUSSION
11. Lines 390-399. This seems to be an argument for increasing LLIN usage. If the results show that LLINs are not appreciably reducing malaria burden, why spend more resources on them? Don’t these results rather support the argument that we need to be looking beyond LLINs (at least those treated with pyrethroids) for malaria control?
12. Line 418. Just “Contrarily…”. Remove the “A”.
13. Lines 470-472. “Housing characteristics such as the presence of screens, open eaves, or proper ventilation…effectiveness of measures like LLINs or IRS”. Is that true? Either provide a citation or link this statement back to data from the Results section that supports it.
14. Lines 511-512. “Housing improve as malaria control intervention can be implemented….” I don’t understand this part of the sentence—I think something is missing. Is it supposed to be “Housing improve[ments]”?
15. Line 526 Remove “and”.
LIMITATIONS
16. Should you include the possibility of bias in the survey---that people, when asked if they sleep under LLINs, overestimate the number of nights they use LLINs?
The paper is very readable, even if not edited; only a very few language issues.
Author Response
Please see the attachment

Reviewer 2 Report (New Reviewer)
The study of Barikissou Damien et al, entitled: Long-lasting insecticide-treated nets combined or not with indoor residual spraying, may not be sufficient to eliminate malaria: a case-control study, Benin, West Africa, is an interesting study as it deals with fundamental issue for malaria endemic countries: malaria control strategies and interventions.
In general, the manuscript is well written (minor editing of English language is required), the research methodology is adequately explained and results are presented in an informative way.
I have made minor corrections, added comments and suggestions on the manuscripts’ body that the authors can find in the attached PDF file.
My major concern is that as authors’ state in the Limitations section, this study was carried out in 2014 and malaria control and elimination strategies in the country have changed substantially since that period (IRS intervention has stopped the last three years and LLINs will be no longer distributed and used). The authors in the present study showed that IRS alone or in combination with LLINs showed greater effectiveness in reducing malaria compared to LLINs alone.
As I am not aware of particular details of the current malaria elimination operational plan in Benin, I believe that the authors should provide adequate explanation for this 9- year delay in publishing this data that nowadays might seem outdated to some people, unless dissemination of these findings could be supportive data for policy and decision makers, for vector control strategies and malaria elimination interventions in the country.

Minor editing of English language required. Some sentences need to be rephrased.
Author Response
"Please see the attachment."

This manuscript is a resubmission of an earlier submission. The following is a list of the peer review reports and author responses from that submission.
Round 1
Reviewer 1 Report
Author determines the role of LLINs combined or not with IRS protected against Plasmodium falciparum infection in Benin. The manuscript tried to answer the effectiveness of LLIN for malaria control. However, the present version of the manuscript need to more details on methadology and proper justification for ineffectiveness of LLIN.
1. When the LLIN was distributed and how it was used need to describe.
2. When the IRS was done and what was insecticide used need to describe.
3. efficacy of insecticides needs to be mentioned.
4 It is very obvious to learn that LLIN does not have impact on malaria caeses which need to justify.
5. On what basis wealth index was categorized need to mentioned.
6. In some table author saying economic status and in other it is wealth status which is confusing.
7. Table 1&2 can be merged similarly 3 & 4
8. data on malaria endemicity of both the sites need to provide.
9. Detail methodology of IRS need to be given
English Language need to improve throughout the manuscript.
Reviewer 2 Report
Results do not clearly illustrate the main objective of the authors in their research, i.e., the sufficiency (or not) of using combinations of LLNS and IRS in the elimination of malaria.
Results shown in the Tables can be considered as raw data, i.e., still to be compiled to clearly demonstrate the objective mentioned above. The six (6) Tables could be resumed into one or two, and, the statistical data shown could be provided in a Supplementary Materials section, if necessary.
And many other issues, e.g., a) lines 156 -157, "Controls were uninfected individual defined as any individual with negative rapid diagnosis test." How RDT negative individuals can be 'Controls'? b) The methodology for the "rapid diagnosis test (RDT)" is not explained, nor a reference is given (line 154).
In addition, the manuscript has lots of room for improvements concerning grammar, typographical errors, and presentation in general.
Author Response
Dear Reviewer,
Many thanks for your valuable comments.
Comments and Suggestions for Authors. Please find responses in the blue colour.
- Results do not clearly illustrate the main objective of the authors in their research, i.e., the sufficiency (or not) of using combinations of LLNS and IRS in the elimination of malaria.
Responses
Here is the summary of the results from the main objective of the study. Please fin the detail in the abstract and the body of the paper.
The objective of this study was to assess whether the national deployment of LLINs, with or without IRS, effectively protected against Plasmodium falciparum infection and uncomplicated clinical cases (UCC) in Benin.
Result 1
In urban where only LLINs were implemented, it was found that sleeping under LLINs did not confer significant protection against malaria infection and UCC when compared to any intervention. However, certain neighborhoods exhibited a notable reduction in infection rates ranging from 65% to 85%.
Result 2
In the rural area, the use of LLINs alone, IRS alone, or their combination did not provide additional protection compared to the absence of any intervention. Comparing LLINs alone to IRS alone and LLINs combined with IRS, it was observed that the latter two options conferred 61% and 65% protection, respectively, against malaria infection.
Conclusion
In conclusion, it is not sufficient of using combinations of LLNS combnied or not with IRS in elimination of malaria.
- Results shown in the Tables can be considered as raw data, i.e., still to be compiled to clearly demonstrate the objective mentioned above.
The six (6) Tables could be resumed into one or two, and, the statistical data shown could be provided in a Supplementary Materials section, if necessary.
Responses
A case-control study design was used. So Cases (infection or clinical cases) were compared to the Control (no infection or no clinical cases). That the reason why row data were highlighted (Cases versus Control for comparison).
Table 1a et Table 1b contain descriptive information.
Only four tables contain results about the sufficiency by using combinations of LLNS combined or not with IRS in the elimination of malaria (Table 2a, Table2b, Table3a, and Table3b).
We clarify again local context of malaria control strategies implementation. See page 3 lines 134 to 142.
The statistical model help to analyze the data and gave the main result of the study. So, these results should appear in the body of the paper.
Our approach requires the use of multiple malaria risk indicators, such as human characteristics and behaviour, LLINs usage, IRS implementation, and housing conditions. These indicators should be analyzed at specific risk areas, such as districts or regions, taking into account the local context then allows for more informed decision-making and effective planning. Recently, the World Health Organization (WHO) has emphasized the importance of micro-stratification of data to better understand and address malaria risks at a local level.
Detail was given in the page 20 lines 607 to 611.
- And many other issues, e.g., a) lines 156 -157, "Controls were uninfected individual defined as any individual with negative rapid diagnosis test." How RDT negative individuals can be 'Controls'? b) The methodology for the "rapid diagnosis test (RDT)" is not explained, nor a reference is given (line 154).
Responses
Lines 156-157 change in the current version. It seems that this issue has been already corrected in the last version of the manuscript.
However, the details and references concerning RDT methodology is described in Page 5 lines 218 to 243.
- Comments on the Quality of English Language
In addition, the manuscript has lots of room for improvements concerning grammar, typographical errors, and presentation in general.
Response
This issue has been already considered in the first review. However, a re-reading of the document has been done.

Reviewer 3 Report
The study described is of interest to those involved in making vector control policies. The study was well conducted, and the results were analysed properly. The significance of the results needs to be highlighted and the discussion needs to be bulked up. The conclusion should be based on the results seen. More details in the attached file.

The weakness in the manuscript is that it requires extensive English language editing. The meaning is sometimes lost due to incorrect word use. This paper would be useful if the paper was to undergo major revised.
Round 2
Reviewer 1 Report
Author have addressed the query carefully and the revised version of the manuscript may be accepted for publication.
Author Response
There are no addditional comments from the reviewer1.
Many thanks
Reviewer 3 Report
Although the manuscript has benefitted from extensive English language editing, the manuscript can benefit from a quick reread to eliminate errors such as that found on line 309 where there is an incomplete sentence.
Author Response
Dear Reviewer,
Many thanks for your valuable comments.
Responses
Correction is done.